# Duplex Probe-Based Fluorescence Melting Curve Analysis for Simultaneous Genotyping of rs1126728 and rs11208257 in the Phosphoglucomutase-1 Gene

**DOI:** 10.3390/diagnostics15182345

**Published:** 2025-09-16

**Authors:** Mikiko Soejima, Yoshiro Koda

**Affiliations:** Department of Forensic Medicine, Kurume University School of Medicine, Kurume 830-0011, Japan; misoe@med.kurume-u.ac.jp

**Keywords:** fluorescence melting curve analysis, phosphoglucomutase-1, rs1126728, rs11208257, dual-labeled fluorescence probe

## Abstract

**Background/Objectives:** Phosphoglucomutase-1 (PGM1) is an enzyme that plays important roles in glycolysis, glycogen metabolism, and glycosylation. The PGM1 gene harbors two common nonsynonymous single-nucleotide variants (rs1126728 and rs11208257), which result in four functional PGM1 phenotypes. Correlations between PGM1 polymorphisms and several pathological conditions have been suggested. **Methods:** To identify the rs1126728 and rs11208257 concurrently, a fluorescence melting curve analysis (FMCA) was developed that utilizes two distinct dual-labeled fluorescence probes. Two distinct Taq polymerases, one with and one without 5′-3′exonuclease activity, were compared. This method was then applied to 95 unrelated Japanese subjects. **Results:** Both Taq polymerases, with and without 5′-3′exonuclease activity, were found to be sufficiently functional. Furthermore, the results of the FMCA using both Taq polymerases were compared with the direct Sanger sequencing results of PCR products from the 95 Japanese subjects, demonstrating 100% concordance. **Conclusions:** The duplex probe-based FMCA developed in this study is useful for examining the association between rs1126728 or rs11208257 and a range of pathological conditions using a relatively large number of subjects.

## 1. Introduction

Phosphoglucomutases (PMGs) belong to the phosphohexose mutase family and fulfill important roles in glycolysis, gluconeogenesis, and glycosylation. These enzymes catalyze the bidirectional conversion of glucose-1-phosphate and glucose-6-phosphate [1]. Currently, five different human PGMs have been documented: PGM1, PGM2, PGM3, PGM4, and PGM5 [2,3,4,5,6]. PGMs have been reported to be polymorphic proteins, and various techniques, such as starch gel electrophoresis, agarose gel electrophoresis, or isoelectric focusing, have been employed to determine the polymorphism of PGMs, especially PGM1, PGM2, and PGM3. In the context of forensic practice, these techniques have historically been employed for the purposes of personal identification and paternity testing because they are Mendelian traits.

In most tissues, PGM1 is the predominant enzyme responsible for 85–95% of the total PGM activity [7]. However, the relative amounts of PGM1 and PGM2 in red cells are approximately equivalent, while PGM3 is not detected [7]. In 1964, two common phenotypes, designated PGM1*1 and PGM1*2, were identified through the application of starch gel electrophoresis [2]. Then, these phenotypes were clearly divided into two subtypes by isoelectric focusing, designated acidic isozyme + (PGM1*1+, PGM1*2+) and basic isozyme − (PGM1*1−, PGM1*2−) [8]. Subsequent molecular genetic analysis indicated that two nonsynonymous single-nucleotide variants (SNVs) (rs1126728; c.661C > T in exon 4; p.R221C, resulting in the 1/2 phenotypes, and rs11208257; c.1258T > C in exon 8; p.Y420H, resulting in the +/− phenotypes) generate four haplotypes [9,10]. One of these is hypothesized to have arisen from multiple homologous intragenic recombination events occurring at a recombination hotspot present between these two SNVs [9,10]. The distribution of these two SNVs across various global populations exhibits a high degree of similarity (dbSNP; https://www.ncbi.nlm.nih.gov/snp/rs11208257#frequency_tab, (accessed on 25 August 2025).

It has been documented that the enzymatic activity of PGM1 in erythrocytes exhibits slight variations among the four phenotypes, with the following order: PGM1*1− < PGM1*1+ = PGM1*2− < PGM1*2+ [11]. Additionally, the enzyme activity of PGM1*2+ expressed in *Escherichia coli* has been reported to be slightly higher than that of PGM1*1+ [12]. In contrast to the enzyme activity of PGM1 in erythrocytes, the enzyme activity of PGM1*1- expressed in *E. coli* was slightly higher than that of PGM1*1+ [12]. Furthermore, the P-21-activated kinase has been demonstrated to bind to, phosphorylate, and enhance the enzymatic activity of PGM1, and the degree of this activation may vary depending on the PGM1 polymorphism [13,14]. The PGM1 polymorphism has been associated with various pathological conditions, including type 2 diabetes mellitus, low birthweight, and repeated spontaneous abortion [14,15,16,17], likely due to slight differences in enzyme activity between polymorphisms, as mentioned above.

Despite the existence of numerous methods for detecting SNVs, probe-based fluorescence melting curve analysis (FMCA) has proven to be a particularly robust approach for the analysis of a relatively large number of subjects [18,19,20,21]. Within a PCR mixture, a non-hybridized fluorescent probe exhibits a weak fluorescent signal at low temperatures. This phenomenon is attributed to the random coil conformation of the probe, which results in fluorescence quenching due to the proximity of the reporter and quencher (Figure 1a). Conversely, the probe exhibits a pronounced fluorescent response when hybridized with its target (Figure 1a). Following the process of dissociation from its target, the probe reverts to a state of reduced fluorescent intensity. The temperature of dissociation, denoted melting temperature (Tm), is observed to decrease in the presence of a mutation [18,19,20,21]. The probe-based FMCA method has been demonstrated to achieve a broad range of melting temperatures, spanning from 4 to 10 °C, through the utilization of a limited number of single-nucleotide variants (SNVs) and diverse fluorescent dyes, thereby facilitating a multiplex assay within a single reaction tube [18,19,20,21].

In the present study, an FMCA method was developed using dual-labeled fluorescence probes for the simultaneous determination of the genotypes of rs1126728 and rs11208257 in *PGM1*. This method may facilitate the execution of association studies of considerable scale between these SNVs and pathological conditions.

## 2. Materials and Methods

### 2.1. Ethical Approval

This research protocol, which used existing anonymized genomic DNA, was reviewed and approved by the Kurume University Ethical Committee (approval number: 22158; approval date: 31 October 2022).

### 2.2. DNA Samples

This study used previously collected and anonymized genomic DNA from 95 unrelated healthy Japanese individuals from Fukuoka, Japan [22]. Unfortunately, only genomic DNA samples were available, and since blood samples were not preserved, PGM1 phenotype analysis could not be performed.

### 2.3. Probes and Primers for a Symmetric Real-Time PCR for Amplification of rs1126728 and rs11208257 in PGM1

Primer3Plus (https://www.bioinformatics.nl/cgi-bin/primer3plus/primer3plus.cgi) (accessed on 25 March 2023) [23] was used to select PCR primers. The forward primer employed for the amplification of a 125 bp fragment containing rs1126728 was rs112678-F: 5′-AGAAGCATCTTTGATTTCAGTGC-3′, and the reverse primer was rs112678-R: 5′-TATCCTCAAGAGATGGGAATTGA-3′. The forward primer for the amplification of a 119 bp fragment containing rs11208257 was rs11208257-F: 5′ AGTGTGGAGGACATTCTCAAAGA 3′, and the reverse primer was rs11208257-R: 5′ CAAGAATTCTCTCTGCCCACTT 3′. A dual-labeled fluorescence probe for the detection of rs1126728 was labeled with 5-carboxyfluorescein (FAM) (5′-FAM-CCGACTGAAGATCCGTATTGATGCT- Black Hole Quencher 1. The position of rs1126728 is underlined). A dual-labeled fluorescence probe for the detection of rs11208257 was labeled with cyanine 5 (Cy5) (5′-Cy5-CATTGGCAAAAGCATGGCCGGAA- Black Hole Quencher 2. The position of rs11208257 is also underlined). The synthesis of all primers and probes was conducted by Eurofins Genomics (Tokyo, Japan). The DNA sequences of the PCR primers, probes, and expected PCR amplification products for rs1126728 and rs11208257 for FMCA are also shown in Figure 1b,c.

### 2.4. Genotyping of rs1126728 and rs11208257 in PGM1 by Sanger Sequencing

Direct Sanger sequencing of all PCR products from 95 Japanese subjects containing rs1126728 or rs11208257 in *PGM1* was performed in a reaction mixture containing 1–10 ng of genomic DNA, 10 µL of Premix Ex Taq (Probe qPCR) (Takara Bio, Shiga, Japan), 250 nM rs112678-F or rs11208257-F, and 250 nM rs112678-R or rs11208257-R with PCR-grade water adjusted to a final volume of 20 µL. The thermal profile comprised the following steps: denaturation at 95 °C for 30 s, followed by 35 cycles of 5 s denaturation at 95 °C, 10 s annealing, and extension at 60 °C. Each PCR primer was used as a sequence primer, as previously described [24].

### 2.5. Genotyping of rs1126728 and rs11208257 in PGM1 by FMCA

Real-time PCR and melting curve analysis were conducted with the LightCycler 480 Instrument II (Roche Diagnostics, Tokyo, Japan). This study compared two premix reagents: Premix Ex Taq (Probe qPCR) and TaKaRa Taq HS Perfect Mix (Takara Bio, Shiga, Japan). The DNA polymerases containing Premix Ex Taq (Probe qPCR) have 5′-3′ exonuclease activity, while TaKaRa Taq HS Perfect Mix lacks this activity. The reaction mixture, with a total volume of 10 µL, comprised 5 µL of 2× either premix, 25 nM to 500 nM each forward primer, 500 nM each reverse primer, 200 nM each dual-labeled fluorescence probe, and 1–10 ng genomic DNA. The same thermal profile was applied to either Premix Ex Taq (Probe qPCR) or TaKaRa Taq HS Perfect Mix: denaturation at 95 °C for 30 s, followed by 45 cycles of 5 s denaturation at 95 °C, 10 s annealing, and extension at 60 °C. Fluorescence data were obtained at the end of each annealing and extension step for both premixes with a combination of FAM (465 nm to 510 nm) and Cy5/Cy5.5 (618 nm to 660 nm) filters.

The probe-based FMCA for both premixes comprised three steps: denaturation at 95 °C for 1 min, hybridization at 40 °C for 2 min, and the acquisition of fluorescence data three times for each temperature range between 50 and 80 °C, with a ramp rate of 0.10 °C/s using FAM and Cy5/Cy5.5 filters. The Tm calling and melting curve genotyping was carried out with the LightCycler 480 Software, version 1.5.0 (Roche Diagnostics), utilizing the default settings. Samples exhibiting analogous melting curve patterns were then systematically consolidated into coherent groups.

## 3. Results

### 3.1. Determination of the Genotypes of rs1126728 and rs11208257 by Direct Sanger Sequencing of PCR Products

Initially, the rs1126728 and rs11208257 genotypes were ascertained by Sanger sequencing of PCR products, which were amplified by forward and reverse primers of rs112678 or rs11208257 in 95 Japanese subjects. For rs1126728, 55 subjects were found to have the C homozygous genotype, 33 exhibited the C/T heterozygous genotype, and 7 were determined to have the T homozygous genotype. For rs11208257, 64 subjects were homozygous for the T allele, 29 subjects were heterozygous for the C/T allele, and 2 subjects were homozygous for the C allele. As shown in Table 1, no deviation from Hardy–Weinberg equilibrium was observed for either SNV (*p* > 0.05).

Six subjects were selected such that each carried one of the following three genotypes at rs1126728 and rs11208257: C homozygote, T homozygote, or C/T heterozygote. Using these six subjects, the optimal primer concentration ratio for asymmetric PCR was determined, and different DNA polymerases were compared.

### 3.2. Determination of the Optimal Primer Concentration Ratio for Asymmetric PCR for Duplex FMCA of rs1126728 and rs11208257

In order to ascertain the optimal primer concentration ratio for asymmetric PCR, the concentration of the rs1126728-R and rs11208257-R primers was fixed at 500 nM, while the concentration of the rs1126728-F and rs11208257-F primers was varied from 25 nM to 500 nM. FMCA was performed using Premix Ex Taq (Probe qPCR), and the results were compared. As shown in Figure 2a–f, the most favorable outcomes for both rs1126728 and rs11208257 were obtained with forward primer concentrations of 25 nM, 50 nM, and 100 nM, regardless of whether the subjects were homozygous or heterozygous for either rs1126728 or rs11208257. However, sufficient FMCA peaks were not observed in forward primer concentrations of 200 nM or 500 nM. Therefore, in subsequent experiments, asymmetric PCR was performed using a forward primer concentration of 50 nM and a reverse primer concentration of 500 nM.

### 3.3. Comparison of Different DNA Polymerases in Probe-Based FMCA to Determine rs1126728 and rs11208257 Genotypes in PGM1

Next, two DNA polymerase premixes were utilized to assess the resolution of duplex probe-based FMCA. The analysis revealed that three genotypes each of rs1126728 and rs11208257 of the *PGM1* could be effectively distinguished when both premixes were used. The employment of Premix Ex Taq (Probe qPCR) or TaKaRa Taq HS Perfect Mix as PCR polymerase resulted in the identification of a single Tm peak at approximately 67.5 °C in the allele homozygous for C/C at rs1126728 (Figure 3a,b). A single Tm peak at approximately 60.0 °C was identified in the allele homozygous for T/T, while double Tm peaks at approximately 60.0 °C and 67.5 °C were identified in the allele heterozygous for C/T (Figure 3a,b). Furthermore, a single Tm peak at approximately 70.5 °C was identified in the allele homozygous for C/C at rs11208257 (Figure 3c,d). A single Tm peak at approximately 63.0 °C was identified in the allele homozygous for T/T, while double Tm peaks at approximately 63.0 °C and 70.5 °C were identified in the allele heterozygous for C/T (Figure 3c,d).

### 3.4. Genotyping of rs1126728 and rs11208257 in 95 Japanese Subjects Using Probe-Based FMCA

Subsequently, the method was applied to genomic DNA samples from 95 unrelated Japanese subjects. A clear distinction was observed among three genotypes of rs1126728 of *PGM1* in all subjects when each premix was used (Figure 4a,b). Furthermore, three genotypes of rs11208257 of *PGM1* were distinctly identified in all subjects (Figure 4c,d). The genotyping results for both rs1126728 and rs11208257, obtained using two premixes for probe-based FMCA, were completely consistent. Additionally, the genotyping results of FMCA were in precise concord with those of direct Sanger sequencing of PCR products from all 95 Japanese subjects. Consequently, both the sensitivity and specificity of the FMCA in this study were found to be 100%.

### 3.5. PGM1 Phenotypes Inferred by Genotyping Results of rs1126728 and rs11208257 in 95 Japanese Subjects

Since only genomic DNA was available for the subjects in this study, it was not possible to perform phenotypic analysis of PGM1. However, the PGM1 phenotype was inferred based on the combination of genotypes for rs1126728 and rs11208257 in 95 Japanese subjects. As demonstrated in Table 2, the PGM1*1/2 or PGM1*+/− phenotypes can be inferred from the genotypes of rs1126728 (C corresponding to PGM1*1 and T corresponding to PGM1*2) and rs11208257 (C corresponding to PGM1*− and T corresponding to PGM1*+). Except when both rs1126728 and rs11208257 were heterozygous, it was possible to identify 8 out of the 10 possible genotype combinations for rs1126728 and rs11208257. This is because either one or both rs1126728 and rs11208257 are homozygous. Therefore, the inferred phenotypes could be determined for these 8 combinations. However, 14 out of 95 individuals were found to be heterozygous for both rs1126728 and rs11208257. Thus, it was not possible to infer whether these 14 individuals were PGM1*1+/PGM1*2− or PGM1*1−/PGM1*2+ using the present method.

## 4. Discussion

As indicated in earlier research, there is a demonstrable association between the PGM1 phenotypes, rs1126728 or rs11208257, and various pathological conditions, including type 2 diabetes mellitus, low birthweight, and repeated spontaneous abortion. PGM1 deficiency was known to be a congenital disorder of glycosylation (CDG) due to several SNVs in *PGM1* [25]. In contrast to the SNVs of PGM1-CDG, the impact of the two SNVs of *PGM1* (rs1126728 and rs11208257) on the encoded proteins was negligible [12]. In vitro, the enzyme activities of the resulting four polymorphic isozymes exhibited a variable order of magnitude, with the following order: PGM1*1− < PGM1*1+ = PGM1*2− < PGM1*2+. The relative activity ranged from 1 to 1.24 [11]. Therefore, these subtle differences may have an impact on certain physiological processes and pathological conditions. Some association studies have employed methods based on the determination of protein phenotypic polymorphism [14,15]. However, such association studies necessitate the implementation of a cost-effective, reliable, and high-throughput genotyping method.

A variety of methodologies were employed to genotype these SNVs in *PGM1* [9,10,17], including direct Sanger sequencing of the PCR product [26], PCR-restriction fragment length polymorphism (PCR-RFLP) [27], PCR-single-strand conformation polymorphism (PCR-SSCP) [28], and hydrolysis probe genotyping (TaqMan) methods [29]. Sanger sequencing is currently considered the gold standard for genotyping SNVs because of its high level of accuracy [30]. However, as with the PCR-RFLP and PCR-SSCP techniques, it necessitates time-consuming and labor-intensive post-PCR procedures, as well as the use of costly reagents. Consequently, it is not well-suited for large-scale analysis [31].

On the other hand, probe-based FMCA does not require additional processing after PCR. Therefore, it reduces the risk of contamination. The dual-labeled fluorescence probe is commonly utilized in a test known as the TaqMan assay. The 5′-3′ exonuclease activity of Taq polymerase causes degradation when the dual-labeled probe hybridizes with a completely complementary target sequence. As a result, the fluorescence intensity of one dye increases as the quenching effect of the other dye is removed [32]. This assay requires two probes to detect a single SNV, one complementary to the wild-type sequence and one to the mutant sequence, which must be labeled with different fluorescent dyes. The dual-labeled probe can be used for more than just hydrolysis probe assays. It can also be used for FMCA. For FMCA, the 5′-3′ exonuclease activity of Taq polymerase is not required, and therefore, an enzyme lacking this activity would appear advantageous as it would not degrade the fluorescent probe. However, in practice, enzymes with 5′-3′ exonuclease activity can also be used. In this study, Premix Ex Taq (Probe qPCR), which contains enzymes with 5′-3′ exonuclease activity, and TaKaRa Taq HS Perfect Mix, which does not exhibit such activity, were utilized without encountering any issues. In either case, asymmetric PCR is required, in which the primer concentration of the complementary strand of the probe is much higher than the primer that amplifies the complementary strand [19]. In this study, the concentration of primers amplifying the complementary strand of the probe required for obtaining favorable results with FMCA needed to be at least five times higher than the concentration of primers amplifying the non-complementary strand (Figure 2a–f). FMCA offers several key advantages, such as the ability to use a single probe for SNV detection and the capability to mix different fluorescent dyes, enabling multiplex assays. In this regard, it outperforms TaqMan assays. In fact, multiplex probe-based FMCA has been used in several studies for other SNVs [19,21,22,33,34,35]. Table 3 summarizes the advantages and disadvantages of single-probe FMCA, double-probe FMCA, and direct Sanger sequencing for genotyping of the two SNVs.

In our previous experiments, only a specific premix (Probe qPCR Mix MultiPlus) produced clear melting curves in triplex assays in probe-based FMCA [35]. In this experiment, however, clear melting curves were obtained with Probe qPCR Mix MultiPlus and several other premixes that had previously failed to generate adequate melting curves for analysis. Consequently, it is hypothesized that a broad array of Taq polymerase premixes can be utilized with the primers and probe combinations employed in this study. Furthermore, the reaction mixtures containing the two types of Taq polymerase utilized in this study exhibit minimal enzyme activation time requirements, and amplification times are also brief. Given that amplification is conducted in two stages, it is hypothesized that the process from PCR amplification to FMCA can be accomplished in approximately one hour. These points are regarded as the present assay’s advantages.

In this study, as shown in Table 2, independent analysis of the rs1126728 and rs11208257 genotypes enabled inference of the PGM1*1/2 polymorphism and PGM1*+/– phenotype polymorphism. If either one or both rs1126728 and rs11208257 are homozygous, the combination of the two genotypes can be determined. However, when both rs1126728 and rs11208257 are heterozygous, it is not possible to infer the combination of the two genotypes, i.e., the haplotype. As a result, haplotype-dependent phenotypic polymorphisms may be either PGM1*1+/PGM1*2− or PGM1*1−/PGM1*2+, and it is not possible to infer either one. This is considered a limitation of this study. However, a major advantage of the present method is that it allows simultaneous analysis of the association between each of the rs1126728 or rs11208257 polymorphisms and the disease using a large number of samples.

## 5. Conclusions

In this study, a probe-based FMCA was developed in order to ascertain the genotypes of rs1126728 and rs11208257 in *PGM1* simultaneously. The present method has the potential to facilitate the association between each of these SNVs and various pathological conditions and traits, including diabetes.

## Figures and Tables

**Figure 1 diagnostics-15-02345-f001:**
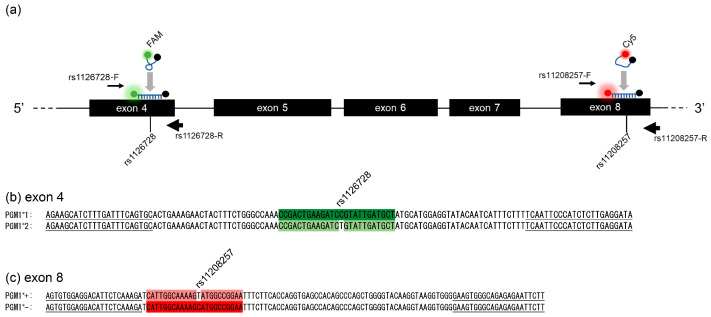
Duplex probe-based FMCA for simultaneous genotyping of rs1126728 and rs11208257 in *PMG1*. (**a**) Relative positions of primers and dual-labeled fluorescent probes. Black boxes indicate *PMG1* exons. Primers are indicated by arrows (reverse primers with higher concentration are shown as thick lines), and dual-labeled fluorescent probes are shown in their respective fluorescent dyes (green: FAM, red: Cy-5). The relative positions of rs1126728 and rs11208257 are indicated by vertical lines. (**b**) DNA sequences of the PCR products for the C and T alleles of rs1126728. The dual-labeled probe sequence for rs1126728 detection is highlighted in green. A mismatched nucleotide in the rs1126728 T allele is shown without highlighting. Primer sequences are underlined. (**c**) DNA sequences of the PCR products for the C and T alleles of rs11208257. The dual-labeled probe sequence for rs11208257 detection is highlighted in red. A mismatched nucleotide in the rs11208257 T allele is not highlighted. Primer sequences are underlined.

**Figure 2 diagnostics-15-02345-f002:**
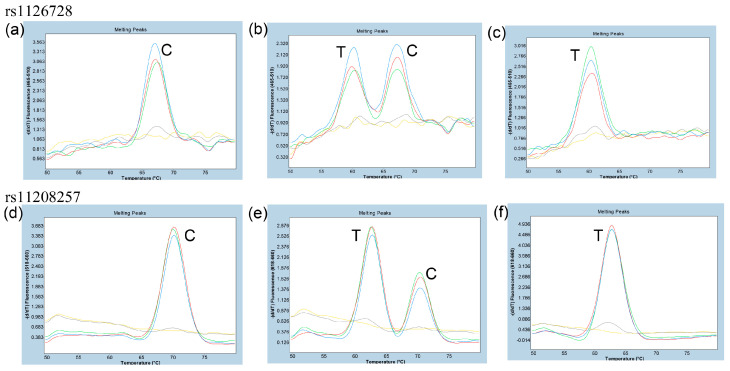
Melting curve patterns of rs1126728 and rs11208257 at various primer concentration ratios in selected Japanese subjects. (**a**) rs1126728 C/C genotype, (**b**) rs1126728 C/T genotype, (**c**) rs1126728 T/T genotype, (**d**) rs11208257 C/C genotype, (**e**) rs11208257 C/T genotype, (**f**) rs11208257 T/T genotype. The concentration of the reverse primers (rs1126728-R and rs11208257-R) was 500 nM, and the concentrations of the forward primers (rs1126728-F and rs11208257-F) were 25 nM (blue), 50 nM (red), 100 nM (green), 200 nM (gray) and 500 nM (yellow).

**Figure 3 diagnostics-15-02345-f003:**
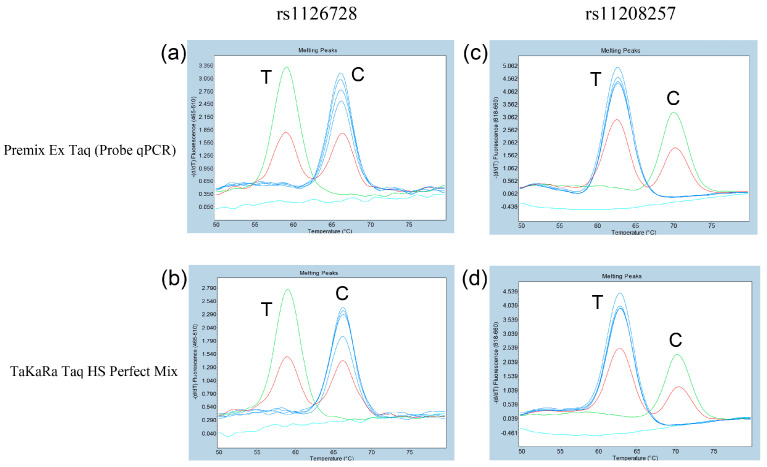
Melting curve patterns to determine the rs1126728 and rs11208257 genotypes of *PGM1* in 6 selected Japanese subjects. (**a**,**b**): Subjects with the rs1126728 C/C genotype (blue), C/T genotype (red), or T/T genotype (green) could be clearly separated. Light blue represents the negative control. (**c**,**d**): Subjects with the rs11208257 C/C genotype (green), C/T genotype (red), or T/T (blue) could also be clearly separated. Light blue represents the negative control. (**a**,**c**) used Premix Ex Taq (Probe qPCR), and (**b**,**d**) used TaKaRa Taq HS Perfect Mix.

**Figure 4 diagnostics-15-02345-f004:**
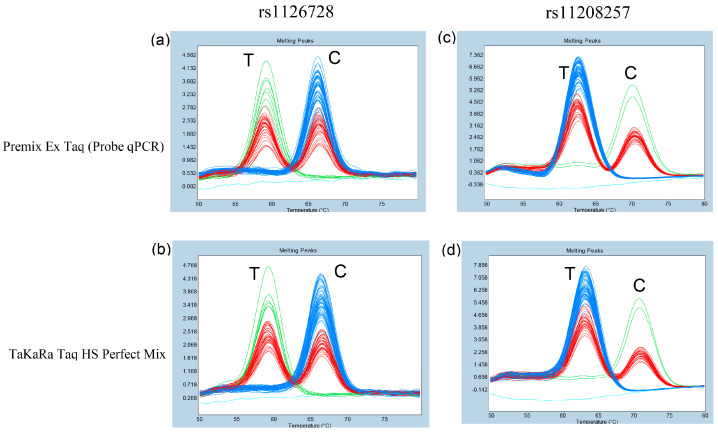
Melting curve patterns to determine the rs1126728 and rs11208257 genotypes of *PGM1* in 95 Japanese subjects. (**a**,**b**): Subjects with the rs1126728 C/C genotype (blue), C/T genotype (red), and T/T genotype (green) could be clearly separated. Light blue represents the negative control. (**c**,**d**): Subjects with the rs11208257 C/C genotype (green), C/T genotype (red), and T/T genotype (blue) could also be clearly separated. Light blue represents the negative control. (**a**,**c**) used Premix Ex Taq (Probe qPCR), and (**b**,**d**) used TaKaRa Taq HS Perfect Mix.

**Table 1 diagnostics-15-02345-t001:** The genotype distributions, allele frequencies, and Hardy–Weinberg equilibrium (HWE) of rs1126728 and rs11208257.

SNV	Genotype	Observed Number	Allele Frequency	Expected Number	Chi-Square
	C/C	55	C = 0.7526 T = 0.2474	53.8132	Χ^2^ = 0.4278 *p* = 0.50
rs1126728	C/T	33		35.3737	
	T/T	7		5.8132	
	T/T	64	T = 0.826 C = 0.173	64.8658	Χ^2^ = 0.3831 *p* = 0.50
rs11208257	C/T	29		27.2684	
	C/C	2		2.8658	

Note: Hardy–Weinberg equilibrium analyses for each SNV were calculated by the chi-square test. A *p*-value greater than 0.05 was considered to be consistent with Hardy–Weinberg equilibrium.

**Table 2 diagnostics-15-02345-t002:** Genotype frequencies of rs1126728 and rs11208257 and PGM1 phenotypes inferred from those genotypes in 95 Japanese subjects.

rs1126728			rs11208257			Inferred Phenotype	Number of Subjects
Melting Temperature	Genotype	Inferred Phenotype	Melting Temperature	Genotype	Inferred Phenotype		
High	C/C	1/1	High	C/C	−/−	1−/1−	1
			Double	C/T	+/−	1+/1−	15
			Low	T/T	+/+	1+/1+	39
Double	C/T	1/2	High	C/C	−/−	1−/2−	1
			Double	C/T	+/−	1+/2− or 1−/2+	14
			Low	T/T	+/+	1+/2+	18
Low	T/T	2/2	High	C/C	−/−	2−/2−	0
			Double	C/T	+/−	2+/2−	0
			Low	T/T	+/+	2+/2+	7

Note: In the phenotype column, 1 represents PGM1*1, 2 represents PGM1*2, + represents PGM1*+, and − represents PGM1*−. Furthermore, 1+ represents PGM1*1+, and 1- represents PGM1*1−. Similarly, 2+ represents PGM1*2+, and 2− represents PGM1*2−.

**Table 3 diagnostics-15-02345-t003:** Comparison between single-probe FMCA, double-probe FMCA, and direct Sanger sequencing for genotyping of the two SNVs.

	Single-Probe FMCA	Double-Probe FMCA	Direct Sanger Sequencing
Time	About × 2 of double-probe FMCA	Fast (approximately one hour)	Time consuming
Cost	About × 2 of double-probe FMCA	Inexpensive	Use of costly reagent
Post-PCR procedures	Unnecessary	Unnecessary	Necessary
Risk of carry-over contaminations	Low probability	Low probability	Relatively high probability
Reproducibility	High	High	High
Accuracy	Accurate	Accurate	Most accurate
Multiple assays	Impossible	Possible	Impossible
Throughput	About half of double-probe FMCA	Relatively high	Low

## Data Availability

Data is contained within the article.

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
