# Peer review of "Duplex Probe-Based Fluorescence Melting Curve Analysis for Simultaneous Genotyping of rs1126728 and rs11208257 in the Phosphoglucomutase-1 Gene"

_diagnostics, 2025, doi:10.3390/diagnostics15182345_

Round 1
Reviewer 1 Report
Comments and Suggestions for Authors
The authors report a modified fluorescence melting curve analysis using duplex probes to genotype two common variants at the PGM-1 gene locus. The manuscript is well written, organized and easy to follow and understand. There are few issues that the authors need to address as follows.
1. Were the subjects healthy and were there phenotypes known. This needs to be clarified in the methods.
2. Estimation of HWE would be a good addition to the results.
3. Lines 190 to 192 are redundant and the same as line 157 to 159.
4. Table 1; What is meant by estimated phenotype? Clarify how was it determined.
5. Line 261: How was the haplotype determined and assessed?
6. It would be valuable to add a table of comparison between single probe, double probe and targeted sequencing for the genotyping of the two variants showing time, cost, reproducibility, accuracy and other relevant information. This would encourage researchers to adapt their protocol for not only these two variants at the PGM-1 loci but for other gene variants.
7. The authors need to clearly state if this method and approach has been used in other studies for other gene variants.
Author Response
Thank you very much for your evaluation of our manuscript. Following your suggestions, we revised our manuscript. Changes were made as follows:
Comments 1: Were the subjects healthy and were there phenotypes known. This needs to be clarified in the methods.
Response 1: Thank you for your comment. Following your suggestion, we have added a description of the subjects' healthy status and unknown PGM1 phenotypes to the "Materials and Methods" section “….healthy Japanese individuals from Fukuoka, Japan [22]. Unfortunately, only genomic DNA samples were available, and since blood samples were not preserved, PGM1 phenotype analysis could not be performed.” (final version: lines 91-93, track changes version: lines 91-93)
Comments 2: Estimation of HWE would be a good addition to the results.
Response 2: Thank you for your helpful suggestion. Following your suggestion, we created new Table (Table 1) about estimation of HWE and added a description ”As shown in Table 1, no deviation from Hardy-Weinberg equilibrium was observed for either SNV (P>0.05).” (final version: lines 164-165, track changes version: lines 165-166)
Comments 3: Lines 190 to 192 are redundant and the same as line 157 to 159.
Response 3: Thank you for your helpful suggestion. Following your suggestion, we deleted redundant sentences (track changes version: lines 197-200, lines 190-192 in previous manuscript).
Comments 4: Table 1; What is meant by estimated phenotype? Clarify how was it determined.
Response 4: Thank you for your helpful suggestion. Following your suggestion, we changed “PGM1 phenotypes inferred from those genotypes” for “estimated phenotype” in a new table (Table 2).
Comments 5: Line 261: How was the haplotype determined and assessed?
Response 5: Thank you for your comment. Following your comment, we have added a explanation of the haplotype determination “Except when both rs1126728 and rs11208257 were heterozygous, it was possible to identify 8 out of the 10 possible genotype combinations for rs1126728 and rs11208257. This is because either one or both rs1126728 and rs11208257 are homozygous”. (final version: lines 240-243, track changes version: lines 245-248)
Comments 6: It would be valuable to add a table of comparison between single probe, double probe and targeted sequencing for the genotyping of the two variants showing time, cost, reproducibility, accuracy and other relevant information. This would encourage researchers to adapt their protocol for not only these two variants at the PGM-1 loci but for other gene variants.
Response 6: Thank you for your helpful suggestion. Following your suggestion, we have added a description (final version: lines 310-312, track changes version: lines 320-322) and created a new table (Table 3) comparing single probe, double probe and targeted sequencing for the genotyping of the two variants.
Comments 7: The authors need to clearly state if this method and approach has been used in other studies for other gene variants.
Response 7: Thank you for your helpful suggestion. Following your suggestion, we have added a description that this method (multiplex FMCA) has been used in other studies for other gene variants (final version: lines 309-310, track changes version: lines 319-320) and have added two references (33 and 34).
Reviewer 2 Report
Comments and Suggestions for Authors
The paper “Duplex probe-based fluorescence melting curve analysis for simultaneous genotyping of rs1126728 2 and rs11208257 in the phosphoglucomutase-1 gene” presents the results of development of FMCA based on duplex probes by authors who, judging by their publications, have been conducting similar studies over the past few years identifying different mutations, genotypes, and different DNA variants based on their unique melting temperatures.
Overall, the manuscript has sufficient scientific validity and contains informative results.
There is a comment on the description of the section (L-229 and further) concerning the results of PGM1 phenotyping. By the way, check the section numbers in the “Results”
Firstly, the reasoning about 8 out of 10 combinations (L-236) is not clearly written.
Secondly, a note should be made in the Table to indicate the numbers "1, 2" and "+ -" that denote phenotypes.
In the Discussion section, the authors write about the limitations of the analysis in relation to heterozygous rs1126728 and rs11208257, but on the other hand, they talk about the potential to facilitate studies of associations of these SNVs with various pathological conditions. It is necessary to clarify whether there is a contradiction here.
Author Response
Thank you very much for your evaluation of our manuscript. Following your suggestions, we revised our manuscript. Changes were made as follows:
Comments 1: By the way, check the section numbers in the “Results”
Response 1: We are sorry about wrong section numbering. Following your suggestion, we corrected the section numbers in the “Results”
Comments 2: Firstly, the reasoning about 8 out of 10 combinations (L-236) is not clearly written.
Response 2: Thank you for your comment. Following your comment, we have added a explanation of the haplotype determination “Except when both rs1126728 and rs11208257 were heterozygous, it was possible to identify 8 out of the 10 possible genotype combinations for rs1126728 and rs11208257. This is because either one or both rs1126728 and rs11208257 are homozygous”. (final version: lines 240-243, track changes version: lines 245-248)
Comments 3: Secondly, a note should be made in the Table to indicate the numbers "1, 2" and "+ -" that denote phenotypes.
Response 3: Following your suggestion, we have added a note to indicate the numbers "1, 2" and "+ -" that denote phenotypes in the new Table 2.
Comment 4: In the Discussion section, the authors write about the limitations of the analysis in relation to heterozygous rs1126728 and rs11208257, but on the other hand, they talk about the potential to facilitate studies of associations of these SNVs with various pathological conditions. It is necessary to clarify whether there is a contradiction here.
Response 4: Thank you for your helpful suggestion. As you pointed out, the present method cannot determine the complete phenotypes that determine the combination of genotypes of two SNVs. However, association studies were conducted for each of the two SNVs in relation to the disease. We have added an explanation of this in the Discussion (final version: lines 240-243, track changes version: lines 275-278) and Conclusion(final version: line 331, track changes version: line 341) sections.